# A History of Fluid Management—From “One Size Fits All” to an Individualized Fluid Therapy in Burn Resuscitation

**DOI:** 10.3390/medicina57020187

**Published:** 2021-02-23

**Authors:** Dorothee Boehm, Henrik Menke

**Affiliations:** Department of Plastic, Aesthetic and Hand Surgery, Specialized Burn Center, Sana Klinikum Offenbach, Starkenburgring 66, 63069 Offenbach, Germany; henrik.menke@sana.de

**Keywords:** fluid management, resuscitation volume, transpulmonary thermodilution, ultrasound, burn resuscitation

## Abstract

Fluid management is a cornerstone in the treatment of burns and, thus, many different formulas were tested for their ability to match the fluid requirements for an adequate resuscitation. Thereof, the Parkland-Baxter formula, first introduced in 1968, is still widely used since then. Though using nearly the same formula to start off, the definition of normovolemia and how to determine the volume status of burn patients has changed dramatically over years. In first instance, the invention of the transpulmonary thermodilution (TTD) enabled an early goal directed fluid therapy with acceptable invasiveness. Furthermore, the introduction of point of care ultrasound (POCUS) has triggered more individualized schemes of fluid therapy. This article explores the historical developments in the field of burn resuscitation, presenting different options to determine the fluid requirements without missing the red flags for hyper- or hypovolemia. Furthermore, the increasing rate of co-morbidities in burn patients calls for a more sophisticated fluid management adjusting the fluid therapy to the actual necessities very closely. Therefore, formulas might be used as a starting point, but further fluid therapy should be adjusted to the actual need of every single patient. Taking the developments in the field of individualized therapies in intensive care in general into account, fluid management in burn resuscitation will also be individualized in the near future.

## 1. Introduction

### 1.1. The Use of Formulas as a Starting Point

Burn patients in the initial resuscitation phase typically require large volumes to restore adequate perfusion pressure and prevent organ failure. To estimate the actual fluid requirements, Baxter and Shires introduced the Parkland formula in 1968 and thus enabled a better outcome of burn patients [1]. Their formula originally proposes a resuscitation volume between 3.5 mL and 4 mL/kg body weight/% TBSA/24 h with half of the fluid volume given early, i.e., in the first 8 h post-burn. Baxter and Shires therefore used several animal studies to determine the decrease in extracellular fluid and fluid loss via the burned surface and subsequently examined the optimal amount of resuscitation fluid as well as the optimal time of fluid administration. Clinically, they showed an adequate urine output using this fluid regimen. In the following years, this study of Baxter and Shires was reduced to the “Parkland formula”, i.e., 4 mL/kg body weight/% TBSA/24 h achieving a urine output of 50 mL/h. However, the second part of the clinical studies of Baxter and Shires is not commonly mentioned in the present discussion on burn resuscitation, and will be discussed later on in this review.

Apart from the Parkland formula, many other formulas have been proposed to estimate the necessary resuscitation volume more closely, e.g., Evans formula [2] or the modified Brooke formula (2 mL/kg/%TBSA) [3]. Those formulas as well as the Parkland formula have been tested in various studies [4,5]. Neither was a more accurate formula found for burn injuries ranging up to 60% TBSA [3] nor has any other formula gained the same popularity as the Parkland formula still has [6,7]. In a recent study with 90 burn patients by Ete et al. [5], the necessary resuscitation volume was 3.14 mL/kg/% TBSA and 3.36 mL/kg/%TBSA for burn patients with concomitant inhalational injury and, therefore, close to the 3.5 mL/kg/%TBSA of the original Parkland formula.

### 1.2. The Hazard of Over-Resuscitation

Though hypovolemia was initially the predominant cause for mortality of burn shock [8] and still increases the risk of acute kidney injury [9], formula-based fluid resuscitation led, in some cases, to hypervolemia, which also provoked adverse effects and increased mortality [10]. In the following years, the symptoms and hazards of over-resuscitation were discussed in the literature. The basic reason for adverse effects of over-resuscitation is the phenomenon of “fluid creep”, which was first described by Pruitt et al. [11]. The excessive fluid does, therefore, not optimize the volumetric status of the patient but rather increases the tissue edema and thus worsens edema-associated complications. Chung et al. stated that the volume given in excess to the initial estimation necessitates even more fluid intake in the following hours [12].

Beside an incline in pulmonary function because of lung edema, the abdominal compartment syndrome (ACS) is one of the most devastating complications with a mortality of over 80% in burn patients [13,14]. Due to hypervolemia and increasing abdominal edema with consequently rising intra-abdominal pressure (IAP), the inferior vena cava (IVC) blood flow and thus cardiac output decrease in first instance with consecutive decreasing urine output and also mechanical ventilation and pulmonary function worsen as early warning signs [15]. Since an increasing abdominal edema and intra-abdominal hypertension (i.e., over 12 mmHg) are not rarely seen, the IAP should be monitored during the resuscitation phase through measurement of the bladder pressure and beyond according to the guidelines of the World Society of Abdominal Compartment Syndrome [16,17].

Still, the discussion about the most suitable method to estimate the adequate resuscitation volumes and how to keep the golden middle of normovolemia is going on. This review will provide an overview of different analyzing methods and parameters to assess fluid requirements. The question of how and when to use colloids or which kind of resuscitation fluid should be used will not be covered by this review. We rather focus on the evolution from a goal-directed resuscitation with the use of different parameters to the most recent development of an individualized resuscitation taking individual organ function into account.

## 2. Early Goal-Directed Resuscitation

To avoid under- and over-resuscitation, different parameters were analyzed in the past to guide fluid management. This “goal-directed” therapy was discussed early in the history of burn resuscitation. As early as 1968, Baxter and Shires used not only urinary output but also a pulmonary catheter (PAC) in the clinical part of their study to measure the cardiac output as an additional parameter [1]. In contrast to a formula-based fluid therapy, the goal-directed approach uses early and regular adjustments of the fluid intake, though a formula-based estimation is also used to start off initially. Thus, the initially estimated fluid intake could be over- or under-estimated in comparison to the given volume. In fact, the comparison of estimated fluid requirements using Parkland or Brooke formula with the real fluid administration showed in several studies an over-resuscitation in 50 to 100% of burn patients [3,18,19,20]. In most cases, the fluid administration was increased to match the goal of adequate urine output (UO).

Therefore, a vital point for sensible goal-directed fluid therapy is which parameters to choose and how to combine them. The different parameters to guide fluid therapy are discussed in this section.

### 2.1. Vital Signs, Urine Output and Serum Lactate/Base Deficit as Parameters

The most widely used parameters are urinary output and vital signs such as blood pressure or mean arterial blood pressure (MAP) and heart rate since they are measurable with minimal effort and in most circumstances and locations worldwide.

Additionally, Baxter and Shires used urine output (UO) as target parameter in their formula-based resuscitation of burn patients. In the following years, urine output remained the leading parameter for goal-directed resuscitation. Though UO can be used easily as an indicator for sufficient resuscitation, it is not a reliable parameter if adequate tissue perfusion and oxygen delivery are the ultimate goals. Furthermore, oliguria or anuria may also be signs for multiple pathologies—not only hypovolemia. In the aforementioned abdominal compartment syndrome, for example, oliguria is a common sign with rising IAP. In fact, UO should not be used as sole parameter to guide resuscitation. However, still it is one of the most popular parameters and still recommended by the American Burn Association [21]. Dries and Waxmann retrospectively compared vital signs and UO with PAC monitoring in their response to fluid administration. Whereas PAC measurements closely reacted after fluid intake with increased cardiac output (CO) and oxygen consumption, vital signs and UO showed no significant change [22]. Saffle et al. demonstrated that UO increased after additional fluid intake with a delay of several hours. Although fluid administration was increased after a decline in UO 8 h after burn, a significant incline of UO was not detectable until 12 h later [23]. At this point, UO peaked up 250 mL/h until 36 h post-burn, though fluid administration was reduced rapidly and thus lagging nearly 12 h behind the initial increase in fluid intake. Therefore, vital signs are not reliable enough and UO as slow reacting parameter inefficient in highly dynamic situations as the burn shock. Hence, these parameters could lead to over-resuscitation when used as goals to guide fluid therapy [11,24].

Of note, serum lactate and arterial base deficit are reliable markers of tissue perfusion. Both parameters indicate uncompensated shock and cell death also in burn patients [25]. However, a rapid lactate clearance shows an adequate resuscitation and subsequently improves survival. As marker of cell death, both parameters rise after the damage of poor organ perfusion is done. Therefore, both parameters should not be used as a goal but rather to confirm adequate resuscitation by rapid correction of initially increased serum lactate and base deficit and to predict a positive outcome.

### 2.2. Static Parameters—CVP and Inferior Vena Cava Diameter

Static parameters, e.g., the central venous pressure (CVP) were often used in the past, though the volumetric status is influenced by many dynamic processes such as changing intrathoracic pressure during in- and expiration with alterations of blood flow and thus changing stroke volume. Therefore, static parameters in general show low reliability to reflect the actual volumetric status of the patient [26]. Multiple studies showed that the CVP does not represent the volumetric status of the patient and also failed to predict the response to fluid intake [27,28].

Accordingly, the sole determination of the diameter of the inferior vena cava (IVC) is not reliable enough and cut-off values depend on the patients’ height. However, the changes in IVC diameter during in- and expiration—i.e., the respiratory variation—proved an adequate tool that can be easily determined via ultrasound bedside [29,30] and will be discussed in Section 3.3.

### 2.3. Thermodilution and Arterial Pressure Wave Analysis

In the last few years, the transpulmonal thermodilution (TTD) has gained popularity as a useful method to determine volumetric parameters as well as cardiac output (CO) and systemic vascular resistance (SVR). The TTD technique (PiCCO©, PulseCO©) uses volume boluses of 10–20 mL of ice cold saline and measures the time between the injection site (central venous catheter) and an arterial catheter (in the femoral or brachial artery). Therefore, it is less invasive than the formerly used pulmonary artery catheter (PAC), which has been replaced by the TTD-technique. TTD enables the calculation of volumetric parameters such as the global enddiastolic volume (GEDV) or intrathoracic blood volume (ITBV) which reliably reflect the actual preload [31].

The increasing pulmonary edema with ongoing resuscitation is represented by the extra-vascular lung water (ELW). Branski et al. could show that an increasing mortality is linked to increased ELW-values [32]. This matches the experience that increasing lung edema prolongs mechanical ventilation as well as length of stay [33]. This parameter is, therefore, recommended as a red flag to stop or reduce further fluid administration.

As a drawback, the TTD- technique and arterial pressure wave analysis presumes a normal cardiac function without cardiac valve dysfunction and excludes patients with arrhythmia. Additionally, a pre-existing history of pulmonary diseases influences the calculation and interpretation of the ELW.

In general, TTD and arterial pressure wave analysis show their greatest reliability not in single measurements, but in detecting changes in the course of ongoing resuscitation.

### 2.4. Thermodilution and “Permissive Hypovolemia”

Initially, goal-directed therapy was used to achieve normal or supra-normal parameters. However, when normal or even supra-normal values of preload parameters (GEDV and ITBV) are targeted, resuscitation volumes estimated by the Parkland formula are exceeded regularly [34,35,36,37]. Surprisingly, no significant change in renal failure, vasopressor use or mortality could be demonstrated in the group of increased resuscitation volumes and with normal preload values in these studies [38]. Thus, optimizing preload parameters to normal or even supra-normal values showed no advantage. As a consequence, Arlati et al. stated that “permissive hypovolemia” was also feasible without the danger of poor organ perfusion and consecutive tissue damage [39]. The goals in this study were defined as a minimum cardiac index (CI = CO/body surface area) of 2 L/min/m^2^ and an hourly urine output of at least 0.5 mL/kg. Arlati and his group showed reduced resuscitation volumes in the first 12 h post-burn, i.e., 3.2 mL/kg/%TBSA, compared to the estimated Parkland formula (4.6 mL/kg/%TBSA). The permissive hypovolemia group even showed an optimized lactate clearance compared to the control group. ITBV as preload parameter ranged between 650 and 750 mL/m^2^ and, therefore, significantly below normal values (900 mL/m^2^). Despite the size of the study (n = 12), Arlati could show significant decreased multiple organ dysfunction scores (MODS) in the permissive hypovolemia group.

In a larger study by Sanchez et al. (n = 132), the cardiac index was also used as primary goal to guide burn fluid therapy [40]. Goals were set as a CI of at least 2,5 L/min/m^2^, ITBV > 600 mL/m^2^ and rapid lactate clearance. The mean volume given was 4.05 mL/kg/%TBSA confirming the results of Charles Baxter. Of note, urine output was not used as resuscitation parameter and the authors reported no correlation between CI and urine output. However, in some cases with a UO of >0.5 mL/kg/h, severe hypovolemia with elevated lactate levels were found and vice versa. Hence, both studies demonstrated the CI to be superior as parameter for goal-directed therapy compared to the widely used urine output or preload parameter alone.

## 3. Individualized Fluid Management

In contrast to the goal-directed therapy, which uses a preset goal for every patient, the individualized approach considers co-morbidities and defines the goals in concordance to the individual organ function of every patient. As a consequence of increasing life expectancy, the average age as well as the incidence of co-morbidities has been rising significantly over the last few decades. Whereas occupational health and safety increased over the last years and occupation associated burns are decreasing likewise, the percentage of elderly and morbid patients rises continuously in developed countries [41].

### 3.1. Cardiac Function and Fluid Responsiveness

In the course of burn resuscitation, cardiac function is a limiting factor for fluid administration. Thus, it is important to understand the pathophysiologic response to fluid challenges. In short, the myocardium is able to optimize its contractility in a certain range, i.e., within the steep part of the Frank–Starling curve. In this optimum range the myocardium is stretched by adequate preload without over-stretching in case of volume overload [42]. Unfortunately, this optimum range is decreased by several cardiac diseases leading to poor fluid responsiveness. In these patients, low CO will not rise after fluid administration but rather deteriorate with increasing resuscitation volume. Thus, the preset goal of normalized CO is achievable for patients with normal cardiovascular responsiveness to fluid challenge. In contrast, further volume administration and increased preload have no benefit in non-responders. The assessment of cardiac function and fluid responsiveness is therefore a cornerstone of individualized resuscitation.

### 3.2. Parameters of Fluid Responsiveness

The transpulmonary thermodilution (TTD) technique as mentioned above can also be used to analyze the arterial pressure wave (PiCCO©, PulseCO©, FloTrac©). The variation of the pulse pressure variation (PPV) and stroke volume variation (SVV) reliably predict the fluid responsiveness with a sensitivity of 80% [43]. Furthermore, the analysis of the arterial pulse curve enables a continuous measurement and promptly demonstrates changes after increased fluid intake. Thus, the combination of TTD and pulse curve analysis (PiCCO©, PulseCO©) not only displays hypovolemia via preload parameters (GEDV and ITBV) but also predicts fluid responsiveness [44]. Unfortunately, arrhythmia, head of bed elevation [45] as well as the settings of mechanical ventilation, low or changing tidal volume, spontaneous breathing and positive end-expiratory pressure (PEEP) especially, strongly influence SVV and PPV parameters [46]. Further parameters of fluid responsiveness are discussed in the following sections.

### 3.3. Point of Care Ultrasound—POCUS

The use of ultrasound as point of care diagnostic is now widely used. The most intriguing points are its rapid availability and non-invasive use. Additionally, the broad spectrum of applications ranges from identification of different causes of shock (hypovolemia versus pulmonary embolism etc.), examination of potential complications or combined injuries (pneumothorax, Focused Assessment with Sonography in Trauma/FAST), lung ultrasound, cardiac function, volumetric status and fluid responsiveness [47]. Lung ultrasound is useful to estimate lung edema which is a stop sign for fluid administration similar to raised ELW values using the TTD technique.

As mentioned above, the diameter of the inferior vena cava (IVC) is an easily assessed but unreliable parameter for the actual volumetric status. However, the respiratory variation of the IVC—i.e., the dynamic changes in diameter during the respiratory cycle—reliably reflects the volumetric status [48]. The respiratory variation can be expressed as:$$\mathrm{IVC}\;\mathrm{variability}=100\;\times \frac{{\mathrm{IVC}}_{\mathrm{maximum}}\;-{\;\mathrm{IVC}}_{\mathrm{minimum}}}{{\mathrm{IVC}}_{\mathrm{mean}}}$$

The cut-off point for the calculated IVC variability lies at 12% and, thus, measurements over 12% are predictive of fluid responsiveness [48].

Though the variation is influenced by ventilator settings and respiration efforts in spontaneously breathing patients [49], a recent meta-analysis by Zhang et al. showed a high specificity of the respiratory variation of the IVC of 87% and 85% in mechanical ventilation and spontaneous breathing, respectively. Additionally, sensitivity in mechanically ventilated patients was reliable with 81% but moderate in spontaneously breathing patients with 70% [50]. Therefore, additional parameters are useful to assess fluid responsiveness, especially in spontaneously breathing patients.

### 3.4. Echocardiography

Focused cardiac ultrasound includes measurement of contractility of the left and right ventricle, stroke volume, valve dysfunction as well as respiratory changes (stroke volume variation) or changes after fluid challenges [51,52]. Therefore, an initial assessment can be done and the responsiveness to fluid administration can be assessed in the course of resuscitation. Focused echocardiography can be done as transthoracic echocardiography (TTE) or transesophageal (TEE) the latter being more invasive and requiring sedation, yet being independent from mechanical ventilation or thoracic (burn) wounds.

Whereas overt hypovolemia can be determined quickly by visual assessment of the left ventricle (“kissing ventricles” as a typical sign), estimation of contractility and fluid responsiveness makes further measurements necessary. Herein, ejection fraction (EF) and stroke volume should be assessed as parameters for left ventricle contractility. The stroke volume (measured as velocity time index of the left ventricular outflow tract) is used to derive cardiac output [53] and measurement of stroke volume variation during the respiratory cycle is equally to the SVV measured by TTD using the same cut-off values [54]. Therefore, a SVV of 12–14% is highly predictive of a positive fluid response, whereas values below 10% reliably identify fluid non-responders [55].

Recently, the contractility of the right ventricle has been focused on since it is crucial for fluid responsiveness. Therefore, the tricuspid annular plane systolic excursion (TAPSE) should be measured as parameter for right ventricular contractility and right ventricle dilatation and flattening of the septum as the “red flags” for fluid administration should not be missed [56]. Echocardiography of the right ventricle also contributes to the interpretation of IVC measurements. Herein, a wide IVC diameter with low respiratory variation suggests a fluid non-responder. In combination with right ventricle dilatation, this constitutes a red flag for any further fluid administration.

### 3.5. Fluid Responsiveness and Fluid Challenge

Additionally, these parameters can be re-assessed after a fluid challenge. Therefore, mini-fluid challenges of 100 mL of colloids are sufficient to certify fluid responsiveness by optimizing cardiac output by at least 10% [57]. Instead of fluid administration, the passive leg raise (PLR) maneuver also enables a fluid challenge of 250–300 mL by auto-transfusion. In contrast to intravenous fluid administration, the volume effect after PLR persists for 20–45 min and is, thus, reversible. When PLR test is conducted correctly (change patient’s position from semirecumbent to supine with legs 45° elevated) and CO or SVV are measured before and 1–2 min after changing the patient’s position, this test is easy to perform and highly reliable. A meta-analysis of Cherpanath and colleagues found a pooled sensitivity of 86% and specificity of 92% with a summary AUROC of 0.95 [58]. Contraindications for PLR maneuver are raised intracranial pressure, intra-abdominal hypertension and lower limb amputations, the last two conditions attenuating the effect of PLR maneuver and therefore leading to unreliable results.

In contrast to single measurements of SVV or cardiac output, the change after volume challenge—nevertheless, via fluid administration or PLR—is independent from ventilator settings, spontaneous breathing or arrhythmia [58]. Thus, the dynamic analysis of fluid responsiveness using PLR/fluid challenge results in more robust measurements and are therefore useful in a variety of clinical settings.

## 4. Discussion—How to Find the Golden Middle

In 1968, Baxter and Shires proposed 3.5–4.5 mL/kg/% TBSA to estimate fluid requirements in burn patients. Still widely used, this formula seems to match the actual fluid requirements adequately in the majority of patients with moderate burn injuries. However in severely burned patients exceeding 60% TBSA, fluid requirements increase disproportionally. Thus, a formula-based resuscitation will predict fluid requirements more inaccurately with rising TBSA. Cancio et al. showed fluid requirements of 6 mL/kg/%TBSA in burn patients exceeding 80% TBSA [3]. In the history of burn resuscitation, fluid administration was adjusted to different parameters as goal directed fluid management. As an easy to handle parameter, urine output is still widely used to guide resuscitation of burn patients. Though the initial goal of 1 mL/kg/h urine output also used by Baxter and Shires was reduced to 0.5 mL/kg/h in the following years, most burn patients seemed to call for higher resuscitation volumes, named as “fluid creep”. However, major burn injuries are highly dynamic situations calling for close adjustment of fluid therapy. Because of the rapid dynamic in the acute phase of severe burns, static parameters such as CVP cannot be recommended. Even slowly reacting parameters such as UO or lactate and base deficit are not reliable enough to guide the actual fluid administration. They should rather be used to verify adequate resuscitation [23].

The development of transpulmonary thermodilution (TTD) enabled the measurement of preload parameters (GEDV and ITBV), cardiac output (CO/CI) and ELW as a parameter of increasing lung edema with limited invasiveness compared to the pulmonary artery catheter (PAC). Herein, recent studies have shown that preload parameters should be used with caution to guide resuscitation in the acute burn shock since normal values are only achievable by significant over-resuscitation. In contrast, cardiac output or cardiac index proved to be reliable to guide fluid administration and enable a closer adjustment of resuscitation to the actual fluid requirements of the patient. Hereby, goal directed resuscitation using cardiac output allows for permissive hypovolemia and concurrently optimized oxygen delivery [39].

Baxter and Shires transferred their findings in different animal experiments to the clinical setting [1]. The first 11 burn patients were resuscitated with 4 mL/kg/%TBSA lactated Ringer solution and their haemodynamics were analyzed, especially CO measured with PAC. Though the loss of extracellular volume was restored in all cases applying the formula of 4 mL/kg/%TBSA, the authors found three groups of cardiac response to fluid therapy. Young burn patients with up to 50% TBSA showed an increasing CO after fluid administration achieving normal cardiac function after 24 h post-burn. In burn patients exceeding 80% TBSA, CO could be restored after 24 h but thereafter showed a continuous decrease unresponsive to fluid administration. As a third group, Baxter and Shires identified burn patients with low CO early after burn trauma without cardiac response to fluid administration. They defined this group as patients “over 45 years of age”. In a subsequent series of 277 burn patients, these different types of cardiovascular response were confirmed. Though it was not the authors’ conclusion, this study demonstrated that not only resuscitation volume and preset goals but also individual organ function, especially fluid responsiveness, are equally important to optimize resuscitation.

Therefore, the goal directed resuscitation should be refined to an individualized resuscitation. Herein, focused echocardiography plays a vital role to assess cardiac function initially and cardiovascular response to fluid administration in the course of resuscitation. Furthermore, the analysis of fluid responsiveness (SVV, IVC variability, CO) demonstrates the potential improvement of increased volume administration beforehand and thus avoids the hazards of over-resuscitation [56]. Moreover, TTD and echocardiography can be used equally to assess SVV and CO but both techniques are complementary concerning the analysis of volumetric status and lung edema. Additionally, point-of-care ultrasound offers a variety of options concerning differentiating the possible causes of shock or enabling ultrasound guided interventions.

## 5. Conclusions

The difference between the goal-directed and individualized approach is an adjustment of fluid therapy to the preset goals in the first case and an adjustment of multiple target parameters to the organ function of the individual patient in the second case. In short, a “one size fits all” approach sets the criteria to be met by the patient, whereas an individualized approach relies on the assessment of cardiac, pulmonary and kidney function in the first step and consecutively defines reliable goals and necessary fluid administration for this patient in the second step. Thus, individualization can be seen as a future development of the goal-directed therapy.

As Kevin Chung stated: “The complex nature of the body’s response to burn injury compounded by the variable response to resuscitation likely makes the starting point almost irrelevant.” [12]. Therefore, the actual discussion should move on from the question about the most exact formula to the question about the most reliable parameters to estimate individual organ function and to define adequate resuscitation.

## Data Availability

Not applicable.

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
