# Peer review of "A History of Fluid Management—From “One Size Fits All” to an Individualized Fluid Therapy in Burn Resuscitation"

_medicina, 2021, doi:10.3390/medicina57020187_

Round 1

Reviewer 1 Report

Excellent review of the history of fluid resuscitation of burned patients. Clearly described every parameter to be use to monitor the fluid requirements in the burned patient. 

Author Response

We cordially thank Reviewer 1 for his encouraging comment.

Reviewer 2 Report

A history of fluid management -From "one size fits all" to an individualized fluid therapy in burn resuscitation

This is nice paper on the history of fluid resuscitation in burns and also quite a lot more.There are only minor items that I liked to address:

Line 115: ”Though, fluid…”
Should be ”Although fluid….”    or change the structure of the sentence

Line 183: “2,5ml/m2” 
Should be “2,5L/min/m2”

Line 262: “an SVV”
Should be “s SVV”

 Line 305: “as the CVP”
Should be “such as CVP

Author Response

We thank Reviewer 2 for his accurate revision of the manuscript. The addressed points (spellings) have been changed in the manuscript accordingly.

Reviewer 3 Report

The authors present a history of fluid management in the resuscitation of burn patients. Sections 2 and 3 provide an overview of goal-directed and individualized fluid management. Section 4 synthesizes the information presented in the previous sections.

Major revisions:

The use of serum lactate and arterial base deficit as markers of tissue perfusion and cell death is more complicated than the authors present. I do agree with the statement that “both parameters should not be used as a goal but rather to confirm adequate resuscitation …”

The title of section 4, “How to find the golden middle” suggests there is something of a compromise to be had by using a combination of goal-directed and individualized resuscitation strategies, but the conclusions seem to advocate for an individualized approach. Please clarify this.

Minor revisions:

Check definitions—for instance, IAP is not defined.

Use of the phrase “on the contrary” is a little confusing and meaning is clearer if that phrase is removed or replaced with “however” [lines 125, 139, 235]

Suggest checking for informal language (eg ‘eyeballing’ in line 255 should be replaced with ‘visual assessment’ or similar.)

Author Response

We thank reviewer 3 for his comments.

1) The use of serum lactate and arterial base deficit as markers of tissue perfusion and cell death is more complicated than the authors present. I do agree with the statement that “both parameters should not be used as a goal but rather to confirm adequate resuscitation …”

Indeed, serum lactate and base deficit are not easy to interpret and to handle. However, both are widely used in (burn) shock therapy and thus, they must be mentioned in this review. Since they're not applicable as goal parameters this section does not discuss these parameters in detail.

2) The title of section 4, “How to find the golden middle” suggests there is something of a compromise to be had by using a combination of goal-directed and individualized resuscitation strategies, but the conclusions seem to advocate for an individualized approach. Please clarify this.

Individualization is a current development in the field of intensive care medicine, e.g. defining different immunotypes in sepsis therapy or assessment of the individual risk for acute renal failure by different biomarkers. This is a new and ongoing progress and thus the individualized approach can be seen as a future development of the goal-directed therapy. An explanation has been added to the manuscript accordingly.

3) Check definitions—for instance, IAP is not defined.

The abbreviation IAP is added at first mention of the  intra-abdominal pressure in the manuscript for better understanding. The grading of intra-abdominal hypertension as well as measurement of the IAP is described in detail in the current guidelines of the WSAC (see references).

4)Use of the phrase “on the contrary” is a little confusing and meaning is clearer if that phrase is removed or replaced with “however” [lines 125, 139, 235]

The phrase has been replaced by "however" as suggested by the reviewer.

5) Suggest checking for informal language (eg ‘eyeballing’ in line 255 should be replaced with ‘visual assessment’ or similar.)

The term has been replaced according to the reviewer.